# Sequencing Antibody Drug Conjugates in Breast Cancer: Exploring Future Roles

Mary Anne Fenton [1,2], Paolo Tarantino [3,4,5,6,†] and Stephanie L. Graff [1,2,*,†]

1   Legorreta Cancer Center, Brown University, Providence, RI 02903, USA; mfenton@lifespan.org
2   Lifespan Cancer Institute, Providence, RI 02903, USA
3   Medical Oncology, Dana Farber Cancer Institute, Boston, MA 02215, USA; paolo_tarantino@dfci.harvard.edu
4   Breast Oncology Program, Dana Farber Brigham Cancer Center, Boston, MA 02215, USA
5   Harvard Medical School, Boston, MA 02215, USA
6   Department of Oncology and Onco-Hematology, University of Milan, 20122 Milan, Italy
*   Correspondence: sgraff1@lifespan.org
†   These authors contributed equally to this work.

**Abstract:** Antibody drug conjugates (ADCs) have emerged as a highly effective treatment strategy across breast cancer (BC) subtypes, including human epidermal growth factor receptor 2-positive (HER2+), hormone-receptor positive (ER/PR+), and triple-negative breast cancer (TNBC). Over the past twenty years, ADCs have undergone relevant evolutions, from target diversity to payload ratio, to linker design, allowing for a progressive increase in their efficacy. From the first-generation ADC, trastuzumab emtansine (T-DM1), approved in 2013 for HER2+ breast cancer, to next generation ADCs such as sacituzumab govitecan and trastuzumab deruxtecan, to emerging ADCs on the horizon, we continue to see unparalleled efficacy compared to traditional chemotherapy. However, each ADC has brought a new cadre of adverse events for clinicians and patients to manage. Importantly, with the development and approval of several ADCs to treat metastatic breast cancer, there are unanswered clinical questions surrounding how to optimally sequence treatment for patients who may be candidates for more than one ADC and, in general, how to treat patients beyond progression on ADCs. From bench to bedside, in this review, we will discuss the pharmacology and current indications for the novel ADCs trastuzumab deruxtecan and sacituzumab govitecan. Highlighting emerging ADCs and ongoing clinical trials, we will anticipate the changes in the breast cancer treatment paradigm. Lastly, we will outline the available data and current approaches for adverse event management and sequencing strategies for ADCs in clinical practice, including proposed mechanisms of resistance.

**Keywords:** breast cancer; antibody drug conjugates; trastuzumab deruxtecan; sacituzumab govitecan; HER2-low; HER2+; datopotamab deruxtecan (Dato-DXd); triple-negative breast cancer; hormone receptor-positive breast cancer; partitumab deruxtecan (HER3-DX); sequencing

## 1. Introduction

Breast cancer (BC) is the most commonly diagnosed cancer and the leading cause of mortality for women globally. It is currently divided into three primary therapeutic subsets based on prognosis, genomics and growth pathways, and types of therapies. Hormone receptor-positive (ER/PR+), defined as ≥1% estrogen receptor (ER) and/or ≥1% progesterone receptor (PR) expression by immunohistochemistry (IHC), accounts for approximately 70% of BC [1]. Therapeutic options for early-stage ER/PR+ breast cancer include adjuvant endocrine therapy, and for clinical or genomic high-risk ER/PR+, neoadjuvant or adjuvant chemotherapy, as well as adjuvant cyclin-dependent kinase 4/6 inhibitors (CDK4/6i). Human epidermal growth factor receptor 2-positive (HER2+) BC (including ER/PR negative and ER/PR+) accounts for approximately 15% of BC [1]. HER2+ BC is defined by ASCO-CAP guidelines as IHC 3+ or IHC 2+ and positive HER2 gene amplification

via in situ hybridization (ISH) [2]. Triple-negative breast cancer (TNBC) is defined as ER/P-negative and HER2-negative and accounts for 10–15% BC [1], and the therapeutic options for this subtype are limited to chemotherapy with or without the addition of checkpoint inhibitor therapy and, in certain cases, PARP inhibitors. These therapeutic approaches are similar in the metastatic or advanced setting, including hormone-directed therapy for ER/PR+ BC and chemotherapy with/without immunotherapy for TNBC.

From 2013 to today, three antibody drug conjugates (ADCs) were approved for the treatment of breast cancer, with the majority being approved for a variety of therapeutic subsets and one approved in the adjuvant setting. As existing ADCs are being explored in earlier lines of therapy and new ADCs emerge in clinical trials, it is important to consider how this changing landscape may influence current and future clinical decision making.

## 2. Current ADCs in Clinical Practice

HER2 overexpression is correlated with poor clinical outcomes despite chemotherapy and endocrine therapy. Based on this observation, the first monoclonal antibody (mAb), trastuzumab, with therapeutic efficacy for HER2-positive breast cancer, was developed. Treating HER2+ BC with trastuzumab in combination with chemotherapy across the neoadjuvant, adjuvant, and advanced breast cancer (ABC) settings improves patient outcomes, including disease-free survival (DFS) and overall survival (OS). Given this success in treating HER2+ BC, trials were conducted among patients with what is now described as HER2-low BC, BC that is HER2 IHC 1+ or IHC 2+/ISH negative. When patients with HER2-low BC were randomly assigned to receive adjuvant chemotherapy or chemotherapy and trastuzumab, results from the phase 3 NSABP B47 trial demonstrated no survival improvement with the addition of trastuzumab in HER2-low early-stage BC [3].

The first ADC for the treatment of breast cancer (and any solid tumor) gained approval in 2013. ADCs are composed of three components: an antibody, which binds the target antigen on the cancer cell; a payload, the chemotherapeutic bound to the antibody; and the linker, the structure binding the antibody and the payload. With target diversity, linkers with variability in stability and cleavage mechanics, and a wide range of potential payloads, it is possible to imagine an endless number of ADCs on the horizon. Furthermore, the drug–antibody ratio is a factor in the efficacy and/or cytotoxicity of each ADC. Ado-trastuzumab emtansine (T-DM1) is a HER2-directed mAb, trastuzumab, covalently bound through a non-cleavable linker to a tubulin-polymerization inhibitor, emtansine, with a drug–antibody ratio (DAR) of 3.5. This ADC served as an incremental step forward beyond trastuzumab chemotherapy combinations for HER2+ BC. In a phase III EMILIA trial of second-line therapy for HER2+ ABC, as compared to lapatinib plus capecitabine, T-DM1 prolonged median OS (30.9 months vs. 25.1 months; hazard ratio [HR] 0.68; 95% CI 0.55–0.85; $p < 0.001$) [4]. Subsequently, among patients with residual disease in the breast/axilla after neoadjuvant therapy, a phase III KATHERINE trial showed that T-DM1, compared to adjuvant trastuzumab alone, improved three-year invasive disease-free survival (IDFS) (88.3% versus 77.0%; HR 0.50, $p < 0.001$) [5]. The side effects of T-DM1, compared with those of trastuzumab, in this trial included higher incidences of thrombocytopenia and peripheral neuropathy. In a subset analysis, benefits across all subgroups were observed, including in the small tumors and ER/PR+ cohort. Of note, KATHERINE did not demonstrate a reduction in the incidence of brain metastasis among patients treated with T-DM1 [6], although subsequent trials have indicated that T-DM1 may have therapeutic efficacy in patients with brain metastasis [7].

The next generation of ADCs emerged with improved linker technology and enhanced DARs with the approval of trastuzumab deruxtecan (T-DXd) in 2019 and sacituzumab govitecan (SG) in 2021.

T-DXd is a HER2-directed monoclonal antibody, trastuzumab, bound via a cleavable linker to a topoisomerase 1 inhibitor payload (DXd) with a high DAR of 8:1 [8]. T-DXd has demonstrated efficacy in animal models [9] in both HER2+ and HER2-low BC, driven by key differences such as the internalization of the ADC prior to undergoing the degradation

of the linker and a membrane-permeable payload, which allows for the release of the cytotoxic to the surrounding tumor, creating a "bystander" effect [10]. It is estimated that when considering both HER2+ and HER2-low BC, some degree of HER2 IHC expression is detectable in 60–70% of BC, as well as in a multitude of other cancer types. Comparatively, the trophoblast cell-surface antigen 2 (TROP2) epithelial antigen is expressed in most solid cancers and 90% of breast cancers, also making this target potentially "histology-agnostic" [11]. SG is a humanized anti-TROP2 mAb bound via a cleavable linker to an irinotecan metabolite payload, SN-38, which is membrane-permeable and also has a high DAR of 7.6:1 [12]. In Table 1, ADC characteristics are compared, including cytotoxic payload, DAR, linker, and antigen, as well as target cytotoxic payload, DAR, linker, antigen, and target.

Results from these next-generation ADCs have led to approvals across different BC subtypes in the metastatic treatment landscape. DESTINY-Breast02 was a phase III trial which included patients with HER2+ ABC who had experienced disease progression on two prior lines of therapy, including T-DM1, providing the initial data for sequencing an ADC after an ADC. As compared to capecitabine with trastuzumab or lapatinib, the patients randomly assigned to receive T-DXd had improved PFS (17.8 vs. 6.9 months, HR 0.36 [95% CI 0.28–0.45]; $p < 0.0001$) and OS (39.2 vs. 26.5 months; HR 0.66 [95% CI 0.50–0.86]). Initial reports from DESTINY-Breast02 exploring translational endpoints such as mechanisms of resistance are starting to emerge. Of note, DESTINY-Breast02 serves as a high-level proof of concept that sequential treatment with ADCs targeting the same antigen can lead to survival benefits in metastatic breast cancer [13].

DESTINY-Breast03, an open-label phase III trial of adult patients with HER2+ ABC who had received a prior anti-HER2 regimen with trastuzumab and a taxane for ABC or who had recurrence <6 months after receiving taxane- and trastuzumab-based adjuvant/neoadjuvant therapy, were randomized to T-DXd or T-DM1, with no prior T-DM1 allowed. Median PFS was 28.8 months for T-DXd and 6.8 months T-DM1 (HR 0.33 [95% CI = 0.26–0.43], nominal $p < 0.0001$), and OS was not reached (NR) for T-DXd (95% CI = 40.5 months–NR) and not reached in the T-DM1 cohort (34.0 months–NR; HR 0.64 [95% CI 0.47–0.87]; $p = 0.0037$) [14].

In DESTINY-Breast02, the most common any-grade treatment-emergent adverse events (TEAEs) with T-DXd included nausea (73%), vomiting (38%), alopecia (37%), fatigue (36%), and diarrhea (27%); the most common grade $\geq$3 TEAEs included neutropenia (8%), anemia (8%), and nausea (7%). Drug-related interstitial lung disease (ILD) occurred in 10% of patients receiving T-DXd, including three patients with Grade-3 ILD, and Grade-2 ILD-associated deaths, as determined via independent adjudication [13]. In DESTINY-Breast03, the most common any-grade TEAEs with T-DXd included nausea (77%), vomiting (52%), alopecia (40%), fatigue (31%), and diarrhea (32%); the most common $\geq$3 TEAEs included neutropenia (16%), anemia (9%), and nausea (7%). ILD occurred in 15% of patients treated with T-DXd, including two patients with Grade-3 ILD and no Grade-4 or Grade-5 events [14]. Notably, 3% of patients treated with T-DM1 developed ILD as well. While there were fewer high-grade ILD reactions with T-DXd, the overall incidence was similar. This suggests increasing provider awareness, enhanced screening, or a decreased incidence among patients treated in earlier lines of therapy may be factors in ILD [15,16]. Early detection, holding T-DXd, and corticosteroids are paramount in effective management [15,16].

The significantly longer PFS seen in DESTINY-Breast03 may represent deeper responses to therapy in earlier lines of disease and in less heavily pretreated disease. However, it may also offer a glimpse into future patterns of disease response in ADC sequencing. Importantly, AEs were similar across the two trials, with no clear evidence that utilizing an ADC after an ADC impacts the toxicity profile of T-DXd.

SG was evaluated first in metastatic TNBC in the phase 1/2 single-group IMMU-132-01 trial and the phase 3 ASCENT trial, and later in ER/PR+ ABC in the TROPiCS-02 trial. In the phase 1–2 basket trial IMMU-132-01, 108 patients with a median of three prior lines of chemotherapy (range 2–10) were enrolled. There was an overall RR (ORR) of 33% (95%

CI = 24.6–43.1), median PFS of 5.5 months (95% CI 4.1–6.3), and median OS of 13 months (95% CI = 11.2–13.7), leading to accelerated FDA approval. TEAEs included four deaths during the trial; Grade-3 or Grade-4 adverse events (in ≥10% of the patients) included anemia and neutropenia, and there was a 9.3% incidence of febrile neutropenia [17].

In ASCENT, a phase III randomized trial for advanced TNBC comparing the effect of SG vs. treatment of physicians' choice (TPC; eribulin, vinorelbine, or gemcitabine) on patients after two or more prior lines of chemotherapy, the patients randomized to SG had a median of three prior lines of therapy (range 1–16). Among patients without brain metastases, SG demonstrated a statistically significant PFS of 5.6 months (95% confidence interval [CI], 4.3 to 6.3; 166 events) versus 1.7 months (95% CI, 1.5 to 2.6; 150 events, [HR 0.41; 95% CI, 0.32 to 0.52; *p* < 0.001), and a median OS of 12.1 months (95% CI, 10.7 to 14.0) vs. 6.7 months (95% CI, 5.8 to 7.7) with TPC [HR (0.48; 95% CI, 0.38 to 0.59; *p* < 0.001)]. TEAEs reported with SG included neutropenia (63%), diarrhea (59%), nausea (57%), alopecia (46%), fatigue (45%), and anemia (34%). Grade-3 TEAEs included neutropenia (51%), diarrhea (10%), anemia (8%), and febrile neutropenia (6%) [18].

For ER/PR+ ABC, SG was compared with TPC (capecitabine, vinorelbine, gemcitabine, or eribulin) in the TROPiCS-02 phase 3 trial. Patients enrolled in this study had an endocrine-resistant ABC that had previously progressed on CDK4/6i and at least two prior lines of chemotherapy. The patients enrolled had a median three prior lines of chemotherapy for ABC, and 95% of patients had visceral metastasis. All patients had prior ET, including a CDKi and in the SG group, and 42% had ≤2 lines of prior chemotherapy compared to 44% for TPC. SG significantly improved outcomes, with a median PFS of 5.5 months versus 4.0 months with TPC (HR 0.65, 95%CI, 0.53–0.83, *p* = 0.0003) and a median OS for SG of 14.4 versus TPC 11.2 months (HR 0.79, 95%CI 0.65–0.95, *p* = 0.020) [19,20]. Grade ≥3 AEs, including neutropenia (52%), diarrhea (10%), and anemia (7%), occurred in 74% of the SG-treated patients; in comparison, Grade ≥3 AEs such as neutropenia (39%), thrombocytopenia (4%), fatigue (4%), and dyspnea (4%) occurred in 60% of the TPC-treated patients.

The differences observed with SG across BC subtypes is likely related to nuances in the study populations and the variability in subtype response rate. Little can be understood about sequencing strategies based on these early reports of SG.

Given early data from both animal models and phase 1 studies, T-DXd moved forward in the DESTINY-Breast04 trial in the newly defined HER2-low BC population. By design, the majority of patients had ER/PR+ ABC (88.7%), and the remainder had tumors that would have previously been described as TNBC (11.3%). DESTINY-Breast04 demonstrated an improvement in median PFS in the ER/PR+ cohort (T-DXd 10.1 months and TPC 5.4 months [HR 0.51]; *p* < 0.001), as well as improved OS (T-DXd 23.9 months vs, TPC 17.5 months [HR 0.64]; *p* = 0.003). In the ER/PR-negative cohort, the median PFS was 8.5 months with T-DXd compared to 2.9 months with TPC (HR 0.46, 95% CI = 0.24 to 0.89); median OS in the cohort also improved with T-DXd (18.2 months vs. 8.3 months, HR 0.48, 95% CI 0.24–0.95). All-grade TEAEs with T-DXd included nausea (73%), vomiting (34%), alopecia (37%), fatigue (48%), and diarrhea (22%); Grade ≥3 TEAEs included neutropenia (14%), anemia (8%), and nausea (5%). Adjudicated drug-related ILD or pneumonitis occurred in 12.1% of patients, including Grade-5 ILD in 0.8% patients [21].

Trials of ADCs in earlier lines of therapy are ongoing across breast cancer subtypes. ASCENT-03 (NCT05382299), a phase 3 study to evaluate first-line SG versus TPC for patients with ABC/TNBC with PD-L1 negative (CPS < 10) or PD-L1- positive tumors (CPS ≥ 10) who received anti-PD-L1 in the neo/adjuvant setting, is open to accrual. TPC includes gemcitabine/carboplatin, paclitaxel, or nab-paclitaxel. In addition, ASCENT-04 (NCT05382286), which aims to investigate the PFS between SG and pembrolizumab versus TPC and pembrolizumab in previously untreated locally advanced ABC/TNBC with PD-L1 expression, is open to enrollment. The DESTINY Breast-06 (NCT04494425) is set to investigate ER/PR+-, HER2-low-, or >0 <1+ endocrine therapy-resistant ABCs and will

involve randomly assigning patients to receive T-DXd or TPC (capecitabine, paclitaxel, or nab-paclitaxel).

**Table 1.** Completed phase III clinical trials involving antibody drug conjugates.

| | Trastuzumab Emantasine (T-DM1) | Trastuzumab Deruxtecan (T-DXd) | Sacituzumab Govitecan (SG) |
|---|---|---|---|
| Target antigen | HER2 | HER2 | Trop-2 |
| Linker cleavage | No | Enzymatic | pH-dependent and enzymatic |
| Membrane-permeable Payload? | No | Yes | Yes |
| hydrophobic | | low | |
| Payload/mechanism of action | Maytansine/ Tubulin inhibitor | Deruxtecan/ Topoisomerase 1 inhibitor | SN-38/ Topoisomerase 1 inhibitor |
| Drug–antibody ratio (DAR) | 3.5:1 | 8:1 | 7.6:1 |
| Phase III Setting | • HER2-positive ABC 2L [4] <br> • Post-neoadjuvant residual disease [5] <br> • HER2-positive ABC 3L [14] | • HER2- positive ABC 2L [13] <br> • HER2-low ABC 2L [21] | • TNBC ABC 2L [18] <br> • ER/PR+ ABC ≥ 2L [20] |
| Common AE | fatigue, nausea, musculoskeletal pain, thrombocytopenia, headache, elevated transaminases, reduction in LVEF | nausea, leukopenia, anemia, thrombocytopenia, elevated transaminases, diarrhea, hypokalemia, cough | neutropenia, nausea, diarrhea, fatigue, alopecia, anemia, vomiting, constipation, decreased appetite, rash, abdominal pain |

Abbreviations: Advanced breast cancer—ABC; Triple-negative breast cancer—TNBC; prior lines of therapy—L; left ventricular ejection fraction—LVEF.

## 3. The Wave of the Future: The Next-Generation ADCs

Given the successes of SG and T-DXd, it is unsurprising to see a progressively rising interest in developing novel ADCs to treat breast cancer. Key examples include the human epidermal growth factor receptor 3 (HER3)-directed ADC patritumab deruxtecan (HER3-DXd), which is bound with a cleavable linker to a topoisomerase I inhibitor payload (DAR 8), as well as the TROP2-directed ADC datopotamab deruxtecan (Dato-DXd), which consists of a humanized immunoglobulin G1 (IgG1) mAb, a cleavable linker, and a topoisomerase-I inhibitor cytotoxic payload (DAR 4).

HER3-DXd results have been reported in clinical trials, including a phase 1/2 study in patients with HER3-expressing ABC (NCT02980341) [22], a phase 2 trial among patients with ABC (NCT04699630) [23], as well a window-of-opportunity trial in early-stage breast cancer [24]. Collectively, the trials showed ORR values ranging from 35% to 45%. Responses were seen across a variety of domains, including among HER3-high- and HER3-low-expressing tumors, PAM50 subtypes, and ER/PR+ and ER/PR-negative disease [22]. In one study, 8% (5/60) of patients had previously been treated with SG, although their specific outcomes have not been reported [23].

In the single-arm phase II ICARUS-BREAST01 (NCT04965766) of ER/PR+/HER2-negative/HER2-low ABC with progressive disease on prior therapy with CDK4/6 ET and one line of chemotherapy, 85 patients were enrolled and treated with HER3-DXd, and 56 patients were available for analysis. Of the 56 patients available for analysis, 16 patients had a partial response, 30 patients had stable disease, and 10 patients had disease progression at 3 months. A Grade ≥3 AE was fatigue (14%), and one patient had Grade-1 ILD [25].

Dato-DXd has been evaluated alone (TROPION-PanTumor01, NCT03401385) [26] and in combination with immunotherapy (NCT03742102) [27]. Dato-DXd, as previously described, is an Anti-TROP2 IgG1mAb Topo1 inhibitor with a cleavable linker and a DAR of 4:1 [28]. In TROPION-PanTumor01, among heavily pretreated patients with advanced TNBC, 32% of whom had received prior Topo1 inhibitor-based ADC therapy, the ORR to Dato-DXd was 32% for all comers. ORR for patients with no prior Topo1 inhibitor-based

ADC was 44% (12/27) compared to 12% ORR (2/17; extrapolated) among those previously treated with topo1-inhibitor-based ADC therapy [26]. This diminished response with similar payload after cytotoxic is an early signal of how we can consider ADC sequencing and similar to our foundational understanding of chemotherapy resistance. The ER/PR+ cohort of TROPION-PanTumor01 also benefited from Dato-DXd, with an ORR of 27% (11/41) [29], but no patients had previously been treated with topo1-inhibitors or ADCs. No ILD was reported in the TNBC cohort, and two cases of pneumonitis were reported in the ER/PR+ cohort [26].

The Dato-DXd ADC is currently in phase III trials for different subsets of ABCs. In metastatic TNBC, TROPION-Breast02 (NCT05374512) randomized patients to receive Dato-DXd or TPC (paclitaxel, nab-paclitaxel, carboplatin, capecitabine, or eribulin mesylate) [30]. For patients with ER/PR+ ABC previously treated with 1–2 lines of chemotherapy, TROPION-Breast01 (NCT05104866) randomized patients to receive Dato-DXd 6 mg/kg IV every 3 weeks or ICC or TPC (eribulin mesylate, capecitabine, vinorelbine, or gemcitabine) [31]. TROPION-Breast01 has met accrual, and a press release from AstraZeneca indicates that there are beneficial effects on PFS for ER/PR+/HER-0, 1+ or 2+ and ISH-negative in this trial, favoring Dato-DXd (https://www.astrazeneca.com/media-centre/press-releases/2023/dato-dxd-improved-pfs-in-breast-cancer.html, accessed on 29 September 2023). We eagerly await the publication of the patient characteristics, results, and safety outcomes.

## 4. Advancing Adjuvant/Neoadjuvant Treatment

Notably, many ADCs are being explored in early-stage breast cancer in an attempt to prevent recurrence and ultimately cure a higher rate of patients with BC. TRIO-US B-12 TALENT (NCT04553770) randomized 58 patients to neoadjuvant T-DXd with or without anastrozole for 6–8 cycles. Of note, five patients did not complete neoadjuvant therapy. At first interim analysis, the rate of pathologic complete response (pCR) was low (one patient randomized to T-DXd alone), the ORR was 75% in the T-DXd alone arm and 63% in the combination arm. The most common Grade $\geq$ 3 TEAEs in Arms A and B, respectively, included diarrhea (3.4%, 3.4%), neutropenia (3.4%, 1.7%), fatigue (1.7%, 3.4%), headaches (3.4%, 1.7%), vomiting (3.4%, 1.7%), dehydration (1.7%, 1.7%), and nausea (3.4%, 0%). Other TEAEs included Grade 2 ILD in one patient (1.7%) which resolved with discontinuation of the therapy, and one death from myocardial infarction after severe GI toxicity (possibly related) [32]. The rate of incompletion and early serious TEAEs signal a need for caution, as ADCs are being explored in early-stage settings and in patients that have not previously been treated via cytotoxic therapy.

Similarly, SG is being explored in early-stage breast cancer. SASCIA (NCT04595565) is a phase 3 trial that has randomized patients with TNBC or ER/PR+ breast cancer and residual disease after primary therapy to receive SG for eight cycles or TPC (capecitabine or platinum-based chemotherapy). The trial's interim safety analysis showed a higher rate of all-grade TEAEs and dose delays with SG but similar rates of dose reductions in both arms [33]. NeoSTAR was a phase II response-guided neoadjuvant trial of SG in localized TNBC with an endpoint of pCR in breast and lymph nodes (ypT0/isN0) in which patients received four cycles of SC, and patients with biopsy-proven residual disease had the option of additional neoadjuvant therapy. In this trial of 50 patients, 98% completed four cycles, 26 patients proceeded to surgery, and pCR to SG alone was 30% [34]. ASCENT-05/OptimICE RD is a phase 3 randomized open-label trial of adjuvant SG and pembrolizumab compared to pembrolizumab with or without capecitabine in patients with TNBC and residual disease after neoadjuvant chemotherapy with the primary endpoint of IDFS [35].

Tropion-Breast03 (NCT05629585) is a phase 3 open-label randomized trial of Dato-DXd with or without durvalumab for stage I-III TNBC with residual disease following neoadjuvant therapy with anthracycline and taxane (with or without carboplatin or pembrolizumab). TROPION-Breast03 will randomize patients into groups receiving Dato-DXd

alone, Dato-DXd in combination with durvalumab, or to investigator's choice therapy, the latter of which includes pembrolizumab, capecitabine, or the two in combination.

As ADCs potentially move into early-phase treatment, the consideration of sequencing will continue to be a challenge. With recurrence after early-stage disease and treatment with an ADC, clinicians will need to consider prior antibody targets and cytotoxic therapy, as well as how time from prior treatment impacts known resistance mechanisms.

## 5. ADCs in Sequence

Given the abbreviated and competing times during which SG and T-DXd were developed, little is known about how to optimally sequence these agents. Patients were enrolled across the five key trials between November 2017 and December 2021, with the first approval creating overlapping indications not occurring until August 2022 [13,14,18,19,21,36]. Summary of current available data for ADCs in sequence is shown in Table 2 and discussed below.

**Table 2.** Completed ADC sequencing trials.

| Trial | Target | ADC1/Payload (MOA) | ADC2/Payload | Setting | Outcome |
|---|---|---|---|---|---|
| DESTINY-Breast02 [13] | HER2 | TDM1/ Maytansine (Tubulin inhibitor) | T-DXd/ deruxtecan (Topoisomerase 1 Inhibitor) | 3rd Line | PFS T-DXd 17.8 months vs. TPC 6.9 months (HR 0.36 $p < 0.0001$) |
| DESTINY-Breast03 [14] | HER2 | TDXd/deruxtecan (Topoisomerase 1 inhibitor) | TDM1/ Maytansine (tubulin inhibitor) | 2nd Line | A total of 35.2% of patients treated with T-DXd later received T-DM1. Given the substantial improvement in OS, the degree of crossover may imply no negative impact on outcomes |
| TULIP HER2+ ABC [37] | HER2 | TDM1 (87%)/ Maytansine (Tubulin inhibitor) | Trastuzumab/ duocarmazine (Active toxin alkylates DNA) | 3rd Line+ | PFS trastuzumab duocarmazine 7.0 months [95% CI 5.4–7.2] vs. 4.9 months TPC (HR 0.64, 95% CI 0.49–0.84; $p = 0.002$) |
| TUXEDO-1 [38] | HER2 | TDM1 (60%)/ Maytansine (Tubulin inhibitor) | T-DXd/ deruxtecan (Topoisomerase 1 Inhibitor) | Previously untreated Brain metastasis | Intracranial response rate 73% (11/15); median PFS 14 months independent of prior T-DM1 |

As the next wave of ADCs begin to appear on the market, exclusion criteria in trial design will create complex questions, ranging from what the optimal sequence of therapy is to what outcome can be expected when patients previously treated with an ADC are challenged with a second or third ADC. Mechanistic differences from payload action to target expression will create variability in understanding these questions. As noted, earlier line therapy may also result in changing dynamics based on time from last exposure when considering resistance and sequencing.

DESTINY-Breast02, as previously described, reported sequential ADC therapy utilizing the same target (HER2) [13] with an extraordinary improvement in PFS (T-DXd 17.8 months versus TPC 6.9 months [HR 0.36 $p < 0.0001$]) for T-DXd after T-DM1. DESTINY-Breast03 reported that 35.2% of patients treated with T-DXd later received T-DM1, and given that a substantial improvement in OS occurred, that degree of crossover can be inferred to have not negatively impacted patients [13]. However, we cannot know the full

impact of administering T-DM1 after T-DXd on outcomes based on these data. Trastuzumab duocarmazine is an ADC with a HER2-targeting antibody bound to a ducarmycin payload composed of a toxin that is activated during intracellular update, which then becomes a DNA-alkylator. In the TULIP phase 3 trial, patients with HER2+ ABC treated with ≥2 previous lines of therapy for ABC or previous T-DM1 for ABC were randomized to receive either trastuzumab duocarmazine or TPC. PFS was noted in favor of trastuzumab duocarmazine in a trial of patients mostly pretreated with T-DM1 and with a median of 4 prior lines of therapy. PFS was 7.0 months [95% CI 5.4–7.2] for SYD985 and 4.9 months for TPC (HR 0.64, 95% CI 0.49–0.84; $p = 0.002$). TEAEs for trastuzumab duocarmazine in this trial included conjunctivitis, keratitis, and fatigue [37].

In a single-institution retrospective analysis of 32 patients, the reported outcomes pertain to the sequential administration of an ADC after an initial administration an ADC. This retrospective analysis included patients with a variety of BC subtypes, including 13 cases of ER/PR+ ABC, 19 case of advanced TNBC, and 22 cases of HER2-low ABC. The median PFS with first-line ADC therapy (ADC1) was 7.55 months, and median PFS with second-line ADC exposure (ADC2) was 2.5 months. ADC1 was a HER2-mAb in 23% and TROP2 in 75% of treatments; ADC2 was HER2 in 40% and TROP2 in 54%. After treatment initiation with ADC2, 53.1% of patients had disease progression at first re-imaging suggestive of resistance to a ADC2. When ADC2 was associated with a change in mAb, target PFS was slightly longer (3.25 months versus 2.3 months, with no target change) [39]. ADC1 included a topoI inhibitor payload in 100% of cases, and ADC2 included a topoI inhibitor in 89% of cases, so analyzing PFS via payload was not feasible. Cross-resistance was present in 67% (8/12) when the mAb was the same and the payload changed from ADC1 to ADC2 and 42% (8/19) when both the mAb and payload changed. The results of this retrospective analysis are intriguing, despite the fact that the analysis was limited due to its small number of patients, meaning that the results need to be confirmed in larger prospective trials with rich translational data collection to help understand the shifting landscape.

In TUXEDO-1, which involved a small cohort of patients with newly diagnosed, untreated HER2+ brain metastases, 15 patients were treated with T-DXd. Among the trial participants, nine (60%) had previously been treated with T-DM1 [38]. The intracranial response rate was 73% (11/15), including two complete responses and nine partial responses, and median PFS was 14 months independent of ER/PR-status or prior T-DM1 [38]. The results of this analysis also suggest that administering an ADC after an ADC can be an effective strategy for CNS metastasis.

Prospective randomized clinical trials to address sequential ADC administration with a switch in targets will soon begin. The Translational Breast Cancer Research Consortium (TBCRC) plans to initiate the Treatment of ADC-Refractory Breast CancEr Dato-DXd or T-DXd (TRADE-DXd) trial, which will include patients with HER2-low ABC, 0–1 prior lines of therapy, and no prior topo-1 inhibitors. Treatment will include ADC1 T-DXd (target HER2) or Dato-DXd (target TROP-1) with crossover to the opposite ADC (ADC2) at progression. TRADE-DXd primary endpoints will be ORR, with secondary endpoints set to include PFS, OS. This trial will primarily answer questions about optimal sequencing and mechanisms of resistance when ADCs with the same payload (e.g., a topoI inhibitor or an exatecan derivative) but different mAb targets are utilized.

Given that both SG and T-DXd have commercial approval in overlapping populations, it is possible that large datasets of real-world evidence will begin looking at outcomes when differing mAb (TROP2 vs. HER2) and modestly different payloads (e.g., a topoI inhibitor or an exatecan derivative vs. SN-38, the active metabolite of irinotecan) are utilized. Additionally, translational science endpoints from the major T-DXd and SG trials reporting any predictors of response or mechanisms of resistance would be valuable in decision making for optimal sequencing.

## 6. Resisting ADCs

Resistance mechanisms to ADCs may include target resistance and payload resistance [40]. To illustrate this point, an intriguing investigation was performed on autopsy and archival tumor tissue obtained from three patients with metastatic TNBC, all treated with SG, including one patient with progressive disease, one with stable disease, and one with an exceptional response (45% tumor regression and response over 8 months, according to the Response Evaluation Criteria for Solid Tumors [RECIST]). Tumor tissue was available from pre-SG and at progression from multiple organ sites for RNA and whole-exome sequencing. In the patient with progressive disease, there was undetectable TROP2 RNA and protein expression, suggesting that primary resistance was mediated by the antibody target. While both the patient with stable disease and the exceptional responder had evidence of TROP2 RNA expression, the exceptional responder had concurrent amplification of the loci that encodes TROP2. At the time of progression for the patient who initially demonstrated an exceptional response, two phylogenetic branches showed resistance developing both via the antibody target (TACSTD2, which encodes TROP2) and payload (TOP1, which encodes the cytotoxic target) [41]. This seminal work defines that resistance via both the mAb target and cytotoxic payload are possible mechanisms and that post-progression analysis can help to determine patterns, frequency, and predictors that will need to be developed to help inform ADC sequencing.

One previously elucidated mechanism of resistance was the downregulation of the targets for ADC binding and payload delivery. With T-DM1, target expression drove efficacy due to less ADC binding at the target and the lack of a non-cleavable linker (resulting in no systemic release of the drug) and a non-membrane-permeable payload to broker the bystander effect when minimal binding does occur. For example, in the phase II trial of T-DM1 after prior HER2-directed therapy by Burris et al., it was revealed that, after central confirmation following randomization, 21 out of 95 patients were HER2-low [42]. Compared to the HER2-positive population, these patients had a lower ORR (4.8% vs. 33.8%) and shorter median PFS (2.6 months vs. 8.2 months) [42]. With advances in drug design, T-DXd is overcoming target downregulation as a mechanism of resistance. The linker in T-DXd is cleavable, allowing the payload to be liberated from the antibody after lysosomal degradation on target binding and allowing for internalization into the intracellular environment; also, as the payload is membrane-permeable, it is able to leach into the surrounding tissue to create the bystander effect.

This impact of target downregulation, however, still seems to have an impact with T-DXd. In DAISY, a phase 2 trial treating patients with ABC across the spectrum of HER2-expression (HER2+, HER2-low, and HER2-0) with T-DXd, median PFS varied by HER2 expression, suggesting that T-DXd efficacy is associated (median PFS 11 vs. 4 months for HER2 IHC 3+ vs. 0, respectively). Higher HER2-0 spatial distribution correlated with no response to T-DXd ($p = 0.0008$), and decreased HER2 expression was seen in 65% of patients who progressed. ERBB2 hemizygous deletions and SLX4 mutations may correlate with decreased response or progression [43].

## 7. Novel Strategies and Combinations

Combining ADCs with other agents may help to delay or prevent resistance. Among the most promising combinations is the combination of ADCs and immune checkpoint inhibitors (ICIs) [44].

ICIs such as anti PD-L1 antibodies permit the enhancement of antitumor response from cytotoxic T-cells via the blockade of inhibitory immune interactions. Current trials of ADC–ICI combinations include BEGONIA (NCT03742102), an ongoing two-part open-label platform study aiming to evaluate the safety and efficacy of durvalumab (D), an anti–PD-L1 antibody, in combination with other novel therapies as a first-line therapy for advanced TNBC. BEGONIA Arm 7 is the combination of D and Dato-DXd; results of the part 2 dose expansion phase of 53 patients at a median 7 months demonstrate an ORR of 73.6% (95% CI 59.7–84.7) irrespective of PD-L1 status. Adverse events primarily

included nausea and diarrhea, and Grade $\geq$ 3 toxicity was noted in 41% of the cohort; one patient had Grade-1 pneumonitis [27]. BEGONIA Arm 6 is D in combination with T-DXd for ER-/PR-negative and HER2-low ABC and was reported after a median follow up of 10.1 months with an ORR 57%. Responses were noted in PD-L1 high and PD-L1 low, and the combination was tolerable [45]. ASCENT-04 (NCT05382286), a first-line trial for advanced TNBC with PD-L1 expression of SG, is comparing the efficacy of pembrolizumab vs. TPC with pembrolizumab.

Another novel approach with ADCs is a bispecific antibody with a unique target on each Fab and alternative payload to circumvent cytotoxic therapy resistance. Zanidatamab zovodotin is a bispecific ADC directed at the HER2 target with Fab specificity of trastuzumab and pertuzumab and a unique payload (auristan). In a phase 1 trial of HER2+ breast cancer and gastric cancer patients with advanced disease, ORR was 28%, adverse events include keratits, alopecia and diarrhea. One patient had dose limiting grade 2 keratitis. No ILD was observed [46].

## 8. Conclusions

Next-generation ADCs have changed the treatment and prognosis paradigm for all subtypes of breast cancer in the advanced-stage setting and are now being evaluated in the early-stage setting to improve cure rates. The therapeutic benefits of contemporary and future ADCs may yet benefit the HER2 ultralow and HER2-0, and some ADCs have been confirmed to have an agnostic histological activity.

Together with prognostic improvements, novel ADCs have produced challenges related to certain side effects, which warrant awareness and efforts for their mitigation. As ADCs shift into earlier lines of therapy, potentially even into the early-stage setting, it will be vital to learn to use ADCs in sequence. This will require high-quality real-world evidence, prospective studies, and large translational analyses to determine key biomarkers for responses, predictors for risks of TEAEs, and mechanisms of resistance to both mAb and payloads. This information will help inspire the next wave of pharmacologic advances in cancer treatment as we aim to deliver the most effective therapy with the least amount of side effects to each individual patient.

**Author Contributions:** Conceptualization, M.A.F. and S.L.G.; writing—original draft preparation, M.A.F.; writing—review and editing, M.A.F., P.T. and S.L.G. All authors have read and agreed to the published version of the manuscript.

**Funding:** This research received no external funding.

**Data Availability Statement:** The data presented in this study are available in this article.

**Conflicts of Interest:** Advisory/Consultant: Seagen, Novartis, Pfizer, AstraZeneca, Eli Lilly and Company, Gilead Therapeutics, Astra Zeneca/Daiichi Sankyo, Genentech; Grant/Research support (to institution): AstraZeneca, Novartis, Astra Zeneca/Daiichi Sankyo; Stock Ownership: HCA Healthcare.

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
