# Peer review of "Sequencing Antibody Drug Conjugates in Breast Cancer: Exploring Future Roles"

_curroncol, doi:10.3390/curroncol30120743_

Round 1

Reviewer 1 Report

Comments and Suggestions for Authors

The  present manuscript shed light on ADCs therapy among all breast cancer subtypes. It summarizes the increasing clinical evidence about fthis class of drugs in terms of activity, efficacy and safety. 

Here are some specific suggestions and comments:

The discussion is well-structured and I found particularly interesting the analysis of the available trials to find preliminary evidence on ADCs sequential treatment and combination in breast cancer. However, due to the limited evidence in clinical trials, it might be interesting to include, more real-world evidence about ADCs sequences, to shed light on actual clinical practice.

It might be nteresting to underline safety data and patients-related outcomes (if available) of the discussed ADCs sequences, to better understand the real impact on quality of life too.

Lastly, consider discussing resistance mechanisms more extensively, to provide a rationale for the preference of a sequence compared to another.

Author Response

Reviewer 1: The present manuscript shed light on ADCs therapy among all breast cancer subtypes. It summarizes the increasing clinical evidence about this class of drugs in terms of activity, efficacy and safety.

Here are some specific suggestions and comments: The discussion is well-structured and I found particularly interesting the analysis of the available trials to find preliminary evidence on ADCs sequential treatment and combination in breast cancer. However, due to the limited evidence in clinical trials, it might be interesting to include, more real-world evidence about ADCs sequences, to shed light on actual clinical practice. It might be interesting to underline safety data and patients-related outcomes (if available) of the discussed ADCs sequences, to better understand the real impact on quality of life too.

Reply: This is early days for the next generation (antibody drug conjugates) ADCs, the two next generation ADC TDXd and sacituzumab have recent FDA approvals, and recent entry into clinical setting. Our review covers all the available evidence available in peer reviewed publications and abstract/congresses within our review as of October 1, 2023. Unfortunately, real-world experiences lag even published clinical trial data, as first you have to have commercial availability and regulatory approval leading to real-world utilization to have that data. Authors present this review in the absence of data to help the oncology community consider these issues and be better equipped for clinical decision making in the absence of data.

Reviewer 1: Lastly, consider discussing resistance mechanisms more extensively, to provide a rationale for the preference of one sequence compared to another.

Reply: Thank you. We added additional data on target down regulation as a mechanism of resistance, highlighting differences between T-DM1 and T-DXd in how bystander effect helps overcome

Reviewer 2 Report

Comments and Suggestions for Authors

The review presents the current view of ADC therapy of the Breast Cancer. Because several experimental trials are currently running, the situation is dynamical and the knowledge will grow quickly. The material is chosen properly and well discussed.

A minor comment would be to include some quantitative outcome measurements to table 1. Also, it would be good to put the table in the begining of the text, to focus attention of the reader to key features of the discussed drugs.

As for table 2, I understand the data is sparse, but it would be good to provide as much numerical outcomes, as possible. Especially interesting here is to compare the response to ADC1 with a response to ADC2 (if possible - restricting response to ADC1 to the group that received ADC2).

As a biophysicist I'd love to see response curve with respect to time spacing between ADC1 and ADC2 application (which would show a recovery of the cell population that was killed by the ADC1 due to binding place (spatial vicinity) and payload sensitivity), but I guess recording of this type of data in human is not possible (maybe there is something related in animal models?).

- some acronyms are not expanded in the initial occurence, I guess examples included DAR and SG. Maybe a list of acronyms would be worth considering - there is really a lot of them.

Author Response

Reviewer 2: The review presents the current view of ADC therapy of the Breast Cancer. Because several experimental trials are currently running, the situation is dynamical and the knowledge will grow quickly. The material is chosen properly and well discussed.

Reply: Thank you for this comment and understanding the complex current environment.

Reviewer 2: A minor comment would be to include some quantitative outcome measurements in table 1.

Reply: Any quantitative data added to Table 1 would result in needing to add numerous additional columns/rows to describe breast cancer subtype, line of therapy, comparator arm). If review and/or editors feel this is beneficial, it would be perhaps easier for readability to add a separate table with outcomes of major phase III ADC trials leading to approved therapies.

Reviewer 2: Also, it would be good to put the table in the beginning of the text, to focus attention of the reader to key features of the discussed drugs.

Reply: This is an interesting point that, we agree, would perhaps enhance the readability. Table 1 is referenced early (page 3) of the manuscript, and we would encourage the editorial/layout team to consider this suggestion, as in fact table placement was completed by journal staff.

Reviewer 2: As for table 2, I understand the data is sparse, but it would be good to provide as much numerical outcomes, as possible. Especially interesting here is to compare the response to ADC1 with a response to ADC2 (if possible - restricting response to ADC1 to the group that received ADC2).

Reply: Agree. That available data is provided in the final column to the extent it is publicly available/reported.

Reviewer 2: As a biophysicist I'd love to see response curve with respect to time spacing between ADC1 and ADC2 application (which would show a recovery of the cell population that was killed by the ADC1 due to binding place (spatial vicinity) and payload sensitivity), but I guess recording of this type of data in human is not possible (maybe there is something related in animal models?).

Reply: Reviewer 2 is invited to collaborate! Yes, this is not available in-vivo, although hopefully as ADC sequencing becomes a pressing issue cell models and xenografts can emerge. But another possible future direction would be HER2-PET (rather than FDG-PET) to determine if antibody-target expression correlates with resistance to antibody but not payload. Again, one of the goals in this review is to highlight all the need that exists.

Reviewer 2: some acronyms are not expanded in the initial occurrence, I guess examples included DAR and SG. Maybe a list of acronyms would be worth considering - there is really a lot of them.

Reply: Thank you. We have double-checked the manuscript for first occurrence abbreviations, as well as created an abbreviation appendix. As for the later, it will again be up to the editorial team if is within the bounds of publication to include.

Reviewer 3 Report

Comments and Suggestions for Authors

This is a well-written review article on the current status of ADC in breast cancer. The authors provide a nice table on various phase III clinical trials. Some minor comments:

1) The title says: "sequencing...." ; however, the article is not about antibody sequencing; I feel the more apt title would be "Role of Antibody Drug....".

2) The authors should include a section of 'components of ADC (such a mAb, linker, payloads etc). This is important as it will connect to section 6 and 7, to discuss futuristic strategies to overcome resistance and new designs.

3) As inflammatory response to ADCs is a major challenge, the authors should include a small subsection on this in the limitations of ADCs.

Comments on the Quality of English Language

N/A

Author Response

Reviewer 3: This is a well-written review article on the current status of ADC in breast cancer. The authors provide a nice table on various phase III clinical trials. Some minor comments: The title says: "sequencing...." ; however, the article is not about antibody sequencing; I feel the more apt title would be "Role of Antibody Drug....".

Reply: The theme of this review is to focus on future directions for ADC sequencing, an area of need that is currently impacting clinical practice/decision making without any data on efficacy/outcomes. We suggested changing the title to: Sequencing Antibody Drug Conjugates in Breast Cancer: Exploring Future Roles

Reviewer 3: The authors should include a section of 'components of ADC (such a mAb, linker, payloads etc). This is important as it will connect to section 6 and 7,

Reply: Thank you for this comment. We have added a general description of ADC components, although unique features of ADCs are introduced more specifically with each drug as it is mentioned.

ADC are comprised of three components—an antibody, which binds the target antigen on the cancer cell; a payload, the chemotherapeutic bound to the antibody; and the linker, the structure binding the antibody and the payload. With target diversity, linkers with variability in stability and cleavage mechanics, and a wide range of potential payloads, it is possible to imagine an endless future of ADCs on the horizon. Furthermore, the drug-to-antibody ratio is a factor in efficacy and/or cytotoxicity of each ADC.

Reviewer 3: …and to discuss futuristic strategies to overcome resistance and new designs.

Reply: Thank you, we have added some additional thoughts on strategies for overcoming resistance.

One previously elucidated mechanism of resistance was downregulation of the target for ADCs binding and payload delivery. With T-DM1, target expression drove efficacy, due to less ADC binding at target, non-cleavable linker resulting in no systemic release of the drug, and a non-membrane permeable payload to broker bystander effect when minimal binding does occur. For example, in the phase II trail of T-DM1 after prior HER2-directed therapy by Burris, et al, revealed on central confirmation after randomization that 21 of 95 patients were HER2 low. As compared to the HER2-positive population, these patients had a lower ORR (4.8% vs 33.8%), and shorter median PFS (2.6 months vs. 8.2 months). With advances in drug design, T-DXd is overcoming target down-regulation as a mechanism of resistance. The linker in T-DXd is cleavable, allowing the payload to be liberated from the antibody after lysosomal degradation on target binding and internalization into the intracellular environment; then, as the payload is membrane permeable, it is able to leach into the surrounding tissue to create the bystander effect. (Ref: Burris HA 3rd, Rugo HS, Vukelja SJ, Vogel CL, Borson RA, Limentani S, Tan-Chiu E, Krop IE, Michaelson RA, Girish S, Amler L, Zheng M, Chu YW, Klencke B, O'Shaughnessy JA. Phase II study of the antibody drug conjugate trastuzumab-DM1 for the treatment of human epidermal growth factor receptor 2 (HER2)-positive breast cancer after prior HER2-directed therapy. J Clin Oncol. 2011 Feb 1;29(4):398-405. doi: 10.1200/JCO.2010.29.5865. Epub 2010 Dec 20. PMID: 21172893.)

3) As inflammatory response to ADCs is a major challenge, the authors should include a small subsection on this in the limitations of ADCs.

Reply: I believe the reviewer means Interstitial Lung Disease (ILD) with T-DXd? We have discussed that as a less common side effect that, due to severity, remains an adverse event of special interest. We have added comment that early detection, holding T-DXd, and corticosteroids remain parmount, citing two great reviews on the subject: PMID 37207309 and PMID 37207306 for readers to have more information.